# The Effects of Exposure to Flupyradifurone on Survival, Development, and Foraging Activity of Honey Bees (*Apis mellifera* L.) under Field Conditions

**DOI:** 10.3390/insects12040357

**Published:** 2021-04-16

**Authors:** Yi Guo, Qing-Yun Diao, Ping-Li Dai, Qiang Wang, Chun-Sheng Hou, Yong-Jun Liu, Li Zhang, Qi-Hua Luo, Yan-Yan Wu, Jing Gao

**Affiliations:** 1Key Laboratory of Pollinating Insect Biology, Institute of Apicultural Research, Chinese Academy of Agricultural Sciences, Ministry of Agriculture, Beijing 100093, China; guoyi63671@163.com (Y.G.); dqyapis@163.com (Q.-Y.D.); daipingli@caas.cn (P.-L.D.); wangqiang@caas.cn (Q.W.); houchunsheng@caas.cn (C.-S.H.); liuyongjun@caas.cn (Y.-J.L.); zhangli3393@126.com (L.Z.); 2Bureau of Landscape and Forestry, Miyun District, Beijing 101500, China; luoqihua0825@163.com

**Keywords:** flupyradifurone, honey bee, pesticide, development, foraging activity

## Abstract

**Simple Summary:**

Honey bees play an invaluable role in ecosystem stability and global food security. Recently, much attention has been directed toward the safety of pesticides to bees. Flupyradifurone (FPF) is a new butenolide insecticide and is considered friendly to honey bee fitness according to risk assessment procedures. Although no significant side-effects on bee colony strength parameters at FPF field-realistic concentration, laboratory experiments have demonstrated that FPF has multiple negative effects on the behavior of individual honey bees. The information suggested that FPF is posing potential risks to honey bees. In this study, we found that the survival rate of bees exposed to FPF was statistically significantly reduced, whereas there were no negative effects on larvae development nor foraging activity. In addition, immune- and detoxification-related genes were upregulated in exposed foragers and newly emerged bees, suggesting that more important synergistic and behavioral effects that can affect colony fitness should be explored in the future.

**Abstract:**

Flupyradifurone (FPF) is a novel systemic nAChR agonist that interferes with signal transduction in the central nervous system of sucking pests. Despite claims that FPF is potentially “bee-safe” by risk assessments, laboratory data have suggested that FPF has multiple sub-lethal effects on individual honey bees. Our study aimed to expand the studies to the effects of field-realistic concentration of FPF. We found a statistically significant decrease in the survival rate of honey bees exposed to FPF, whereas there were no significantly negative effects on larvae development durations nor foraging activity. In addition, we found that the exposed foragers showed significantly higher expression of ApidNT, CYP9Q2, CYP9Q3, and AmInR-2 compared to the CK group (control group), but no alteration in the gene expression was observed in larvae. The exposed newly emerged bees showed significantly higher expression of Defensin and ApidNT. These results indicate that the chronic exposure to the field-realistic concentration of FPF has negligible effects, but more important synergistic and behavioral effects that can affect colony fitness should be explored in the future, considering the wide use of FPF on crops pollinated and visited by honey bees.

## 1. Introduction

In recent decades, many insect pollinators have consistently declined in population and biodiversity, posing a potential threat to the existence of species and global food security [1]. More specifically, there is a serious concern about the decline of the western honey bee (*Apis mellifera*) worldwide. The global honey bee populations are in sharp decline as driven by a combination of factors such as parasites, introduction of invasive species, shrinking food sources, and persistence of chemical residues [2,3,4,5,6]. One of the major causes of such a decline is the massive use of pesticides in agriculture. With the widespread use of pesticides in agricultural and horticultural crops, the risk of honey bees coming into contact with pesticides is increasing.

Recently, much attention has been directed toward the safety of pesticides to bees. Risk assessment procedures established by regulatory agencies are designed to ensure that the new approved pesticides are compatible with the protection of bees [7]. Traditional estimates of pesticide safety to bees only consider the lethal level (LD_50_). However, the partial assessment of pesticides cannot fully prove safety to bees. Actually, bees are normally exposed to “field realistic” concentrations, which are far below the lethal dose. Bee scientists and keepers have consistently claimed that there is a correlation between the existence of colony collapse disorder (CCD) and the propinquity of hives to crops that were applied with pesticides [8,9,10]. Agonists of nicotinic acetylcholine receptors (nAChRs) act as systemic pesticides (e.g., neonicotinoids) to interfere with the receptor signal transduction in the target pests’ brain, resulting in paralysis and mortality of target pests [11]. Pesticides can contaminate the whole colony due to the foraging behavior of honey bees in farmland. Growing evidence has suggested that neonicotinoid insecticides have a series of negative effects on honey bees (e.g., climbing ability, olfactory learning and memory, foraging activities, and reproduction [12,13,14]), even at very low doses. In response to the concerns about the many evidences of negative impacts on honey bees, the European Union has banned three highly toxic neonicotinoids (clothianidin, imidacloprid, and thiamethoxam) for outdoor use since 2018 [15]. Nevertheless, agonists of insect nAChRs are still one of the world’s largest selling insecticides because they are not only effective, but also show a noticeably favorable safety to humans and mammals [16].

As the application of neonicotinoid insecticides has been gradually restricted, participants in the pesticide industry are actively developing alternative products with lower environmental impacts. Flupyradifurone (FPF, Sivanto^®^) is a new butenolide insecticide developed by Bayer. Similar to neonicotinoids, FPF is a systemic nAChR agonist that interferes with signal transduction in the nerve center of sucking pests [17]. Compared with neonicotinoids, FPF has a lower binding affinity to insect nAChRs, but is effective against many pest insects that are resistant to the neonicotinoid [18]. Recent data indicate that bees forage and eat less nectar containing FPF [19]. This may imply that bees can repel FPF through their taste or olfactory system, and this character makes the FPF relatively safe for bees. The US Environmental Protection Agency has declared that FPF appears to have a favorable safety profile for honey bee colonies [20]. Under field conditions, no adverse effect on the number of adult bees, larvae, and pupae in the colony as well as the weight of honeycomb and brood chambers after exposure to FPF [21].

While laboratory experiments have demonstrated that FPF has multiple detrimental effects on the behavior of individual honey bees. It has been demonstrated that exposure to FPF impaired olfactory associative function in honey bees. After exposure to a high concentration of FPF, the learning performance was significantly decreased in *A. mellifera* [22,23,24]. For *Apis cerana*, chronic exposure to FPF impaired olfactory learning in both larval and adult bees, and bee larvae were more sensitive to FPF [25]. It has been demonstrated that a sub-lethal dose of FPF induced a significant reduction in adult emergence [26]. In addition, interaction with other environmental stressors may well make FPF more detrimental to honey bees than FPF examined as a single factor. Under nutritional stress, exposure to a field-realistic dose of FPF can decrease survival rate, food consumption, flight ability, and the thermoregulation of honey bees [27]. The combination of FPF and a common fungicide (propiconazole) increased abnormal behaviors of both in-hive and forager bees even at FPF field-realistic concentration (worst-case scenarios) [28]. Co-exposure with FPF and a bee parasite *Nosema ceranae* can increase mortality and alter the expression of immune and detoxification related genes in honey bees [26]. Taken together, the information suggests that FPF poses a potential risk to honey bees, despite FPF being considered as “bee safe” by the EPA. Thus, further detailed experiments are required to detect the negative impacts of field-realistic concentrations of FPF.

The purpose of our study was to expand the EPA studies to the effects of field-realistic concentration of FPF in the following aspects: the growth and development of larvae; the homing and pollen collection ability of forager bees; and the expression of genes related to cellular immune, detoxification, and division in larvae, newly emerged bees, and foragers. This research will provide additional information of FPF exposure risks to honey bees, and ultimately help determine if the honey bee is protective.

## 2. Materials and Methods

### 2.1. Colonies

Experimental colonies (*Apis mellifera*) were maintained at an apiary in the Institute of Apicultural Research (40°00′43″ N, 112°12′43″ E), Chinese Academy of Agricultural Sciences (Beijing, China). Each hive consisted of six frames of adult bees, one empty frame for brood, and an in-hive frame feeder. For the field experiment, we used eight healthy and strong colonies with no history of pesticide exposure or other bee diseases, four to the FPF treatment group and four to the control group. Each hive contained approximately 2 kg honey bees.

### 2.2. Flupyradifurone (FPF) Exposure

Flupyradifurone (purity 99.5%) was purchased from ChemService (West Chester, PA, USA). Stock solution was prepared by dissolving the powder in MilliQ-water and then diluted with sucrose solution (50% *w*/*v*). The final concentration of 4.0 ppm FPF was applied to the following experiments. This concentration corresponds to the nectar brought back by bees foraging on oilseed rape treated with the recommended FPF concentration (4.3 ppm and 4.1 ppm) [20]. The sucrose solution for control groups was identical but FPF free. Colonies were fed 1 kg of control or FPF sucrose solution with a black plastic in-hive frame feeder after the eggs began to hatch (first instar larvae stage). All colonies were fed with the syrup for six days, which covered the whole larval stage. This procedure mimicked the natural situation, where bees also had the opportunity to forage freely in natural food sources. Thus, the experiments were conducted from August to September when the nectar flow period had already passed. The amount of sucrose solution represents the major fraction of what colonies might gather and process over such a time period.

### 2.3. Growth and Development of Larvae

To quantify the influence of FPF on larval development, honey bee queen in each colony was restricted onto a caged empty comb for 24 h to allow them to lay eggs. Then, the comb was taken out and covered with a transparent plastic film. The position and number of bee eggs (100–150) were marked on the plastic film. Then, the comb was put into a plastic cage where the queen bee could not enter but the worker bees could, and then put it back into the beehive. The number of honey bee eggs, larvae, pupae, adults or dead bee in the cells were observed and recorded every 12 h. If the cell was newly emptied, the larvae and pupa were considered dead. [26]. The developmental time from egg-hatching until emergence was also recorded for each marked bee.

The developmental durations were calculated as follows [29,30]:
Larval development time = date of pupation − hatching date.Pupal development time = emergence date − date of pupation. Total development time = emergence date − hatching date. 

### 2.4. Foraging Activity 

The foraging activity measurement was performed according to the methods reported before with minor modifications [31]. The foraging activity of the bees were recorded by a video camera at the entrance of the beehive on days 0, 8, and 12 after FPF exposure. The foragers were estimated by counting all bees entering into their hives and those entering into their hives loaded with pollen grains on their legs for 20 min during 7 to 9 a.m. 

### 2.5. RNA Isolation and qRT-PCR Gene Expression Analysis

For qRT-PCR (Quantitative Reverse Transcription PCR) analysis, the fifth instar larvae, newly emerged bees were sampled respectively, meanwhile foragers were caught randomly at the hive entrance on the last day of exposure. Ten bees were randomly selected and pooled together as one biological replicate samples (three samples from each colony and four colonies in each treatment). Samples were frozen and stored at −80 °C until the time of RNA isolation. Total RNA was isolated using TRIzol reagent following the manufacturer’s instruction. The total RNA was treated with DNAse-I (Fermentas, Inc6 and purified with an RNeasyMini Kit (Qiagen). RNA quality was determined using the NanoDrop 2000 and RNA quantity was evaluated by an Agilent 2100 RNA Nano 6000 Assay Kit (Agilent Technologies, Inc., Santa Clara, CA, USA). First-strand cDNA was synthesized from 1 mg of total RNA using the Superscript II Kit (Invitrogen, Carlsbad, CA, USA) with oligo d(T)18 primers according to the manufacturer’s instructions. Quantitative RT-PCR reaction contained 100 ng cDNA, 1 pmol of each primer, and 2x Sybr Green PCR buffer (Bio-Rad, Hercules CA, USA) in a final volume of 20 μL. The PCR conditions for the amplification were as follows: 1 min at 94 °C, followed by 30 cycles of 45 s at 94 °C, 60 s at 54 °C, and 75 s at 72 °C. qRT-PCR was performed in triplicate reactions, with the same cDNA pool. The relative expression was calculated against that of the house keeping gene β-actin. Primer pairs used for each gene are listed in Table 1. 

### 2.6. Statistical Analysis

Statistical analyses were performed using SPSS 26.0 Statistical software. Overall survival curves were compared using a Kaplan–Meier survival analysis. The general linear models were applied for comparing responses between experimental groups for developmental durations, and mean numbers of foraging workers entering their colonies and entering their colonies loaded with pollen. The mean number data were log-transformed for the calculations, before the general linear models. ANOVA was used to compare gene expression (qRT-PCR).

## 3. Results

### 3.1. FPF Exposure Reduced the Survival Rate of A. mellifera

In this study, we first investigated the effects of FPF on the survival rate of all the marked *A. mellifera* larvae. There was a statistically significant decrease in the survival rate of honey bees from hives exposed to FPF compared to those from the control hives (*F* = 16.108, *p* < 0.001) (Figure 1). 

### 3.2. No Effects of FPF on the Developmental Duration of Honey Bees

As shown in Figure 2, exposure to FPF did not affect the larval time (*F* = 1.615, *p* = 0.251), pupal time (*F* = 0.272, *p* = 0.621), and both combined (total) (*F* = 0.27, *p* = 0.62) (Figure 2). 

### 3.3. No Effect of FPF on Foraging Activity

We evaluated the foraging activity by accounting the mean number of foraging workers introduced to their hives and bees entering hives loaded with pollen grains on days 0, 8, and 12 after FPF exposure. The data in Table 2 showed that there was no significant difference in foraging activity between the FPF exposed and control group from 7 to 9 am on all of the inspection days. Foragers entering their colonies on day 0: *F* = 0.238, *p* = 0.643; on day 8: *F* = 3.23, *p* = 0.122; on day 12: *F* = 0.479, *p* = 0.515. Foragers entering loaded with pollen on day 0: *F* = 0.2, *p* = 0.67; on day 8: *F* = 1.879, *p* = 0.219; on day 12: *F* = 0.224, *p* = 0.653. 

### 3.4. Gene Expression of Larval after Exposure to FPF 

In this study, we investigated the effects of FPF on the expression of immune genes (Abaecin, Defensin, ApidNT), detoxification genes (cyp9q1, cyp9q2, cyp9q3) in the fifth instar larvae. As shown in Figure 3, there was no significant difference in the expression of test genes between the exposure and control group (Abaecin: *F* = 1.768, df = 3, *p* = 0.232 > 0.05; Defensin: *F* = 0.652, df = 3, *p* = 0.45 > 0.05; Apid NT: *F* = 0.913, df = 3, *p* = 0.376 > 0.05; cyp9q1: *F* = 0.424, df = 3, *p* = 0.539 > 0.05; cyp9q2: *F* = 0.195, df = 3, *p* = 0.674 > 0.05; cyp9q3: *F* = 0.1, df = 3, *p* = 0.763 > 0.05.). 

### 3.5. Gene Expression of Newly Emerged Bees and Forager Bees

The relative expression of immune genes (Abaecin, Defensin and ApidNT), detoxification genes (CYP9Q1, CYP9Q2 and CYP9Q3), foraging gene (Amfor), and division related genes (AmOctαR-1, AmInR-2) in newly emerged bees and forager bees were quantified (Figure 4). Relative expression of Defensin (new bees: *F* = 1.02, df = 3, *p* = 0.035 < 0.05; forager bees: *F* = 0.368, df = 3, *p* = 0.566 > 0.05) and ApidNT (new bees: *F* = 0.898, df = 3, *p* = 0.038 < 0.05; forager bees: *F* = 1.017, df = 3, *p* = 0.003 < 0.05) were both upregulated in newly emerged bees after FPF exposure in comparison with the CK group. In addition, the exposed foragers showed significantly higher expression of ApidNT, CYP9Q2 (new bees: *F* = 0.415, df = 3, *p* = 0.543 > 0.05; forager bees: *F* = 26.432, df = 3, *p* = 0.002 < 0.05), CYP9Q3 (new bees: *F* = 0.377, df = 3, *p* = 0.562 > 0.05; forager bees: *F* = 0.341, df = 3, *p* = 0.001 < 0.05), and AmInR-2 (new bees: *F* = 0.158, df = 3, *p* = 0.704 > 0.05; forager bees: *F* = 2.566, df = 3, *p* = 0.002 < 0.05) compared to the CK group. There was no significant difference in Abaecin (new bees: *F* = 0.57, df = 3, *p* = 0.479 > 0.05; forager bees: *F* = 0.215, df = 3, *p* = 0.695 > 0.05), CYP9Q1 (new bees: *F* = 0.328, df = 3, *p* = 0.588 > 0.05; forager bees: *F* = 0.959, df = 3, *p* = 0.365 > 0.05), Amfor (new bees: *F* = 0.501, df = 3, *p* = 0.505 > 0.05; forager bees: *F* = 1.024, df = 3, *p* = 0.351 > 0.05), and AmOctαR-1 (new bees: *F* = 0.42, df = 3, *p* = 0.541 > 0.05; forager bees: *F* = 3.377, df = 3, *p* = 0.116 > 0.05) expression between the exposed bees and control bees.

## 4. Discussion

Flupyradifurone (FPF) (Sivanto™) is a newly developed systemic insecticide as an alternative to neonicotinoids. Although it has been declared that FPF has a favorable safety profile for honey bees and other pollinators, a meta-analysis of systematic literature studies of potential sub-lethal effects of FPF has demonstrated that exposure to field-realistic levels of FPF exhibited significant sub-lethal impacts on honey bees [39]. This study investigated the effects of FPF on the development on larvae as well as foraging ability of foragers who consume the syrup during exposure. Furthermore, it also provides new insights into the characteristics of genes related to immune, detoxification, and division when exposed to butenolide insecticide.

Results showed that chronic exposed to FPF for six days during larval stage significantly reduced the survival rate of the larvae of *A. mellifera* (Figure 1), which was in line with the previous study reported by Al Naggar et al. [26]. However, Campbell et al. demonstrated no colony-level impacts of FPF on the number of bees, eggs, brood, or colony weight [21]. However, laboratory experiments have demonstrated that FPF exposure not only decreased honey bee larval survival, but also reduced adult emergence [25,26], which could have a chain reaction to colony fitness. Additionally, FPF treatment did not significantly affect the developmental duration of larvae (Figure 2). Our experimental setup aimed to mimic the natural rearing conditions where the FPF contaminated nectar was stored and used for rearing larvae, which means that the developing larvae are continuously fed with contaminated food. The reduced brood survival in pesticide-exposed colonies here could result from impaired nursing behavior (insufficient care for the brood) and temperature control by nest workers [40,41,42]. More work is required to quantify the effects of FPF induced mortalities and whether they can impact their performance and reproductive success after long-time exposure.

Besides mortality and development being direct indicators of insect fitness, many of the upstream effects of pesticide exposure may be behavioral such as foraging performance. Previous studies have implied that the efficiency of collecting pollen by pesticide-exposed bumblebees was significantly lower than the untreated group [43]. In addition, exposure to imidacloprid or clothianidin induced a lower foraging activity and longer foraging bouts in honey bees [44]. We found that there was no significant difference in the frequency of collecting pollen between FPF-exposed bees and unexposed controls (Table 2). The foraging behavior is influenced by both the physiological functions (e.g., flight) and also cognitive mechanisms involved in olfactory functions. According to laboratorial data, FPF exposure can impair sucrose responsiveness and the olfactory learning ability of *A. mellifera* but only at high dosages [22,45]. Recently, Hesselbach et al. demonstrated that chronic exposure to FPF reduced motor abilities and taste and appetitive learning performance of honey bees foraging for pollen and nectar [23,24]. Multiple parameters related to the foraging behavior in FPF-treated colonies are still required for further study.

Studies have suggested that sublethal concentration of pesticides would cause transcriptional alterations involved in immune and detoxification functions in bee species [46,47]. The response of detoxification-related genes (such as cytochrome P450) and antimicrobial peptides [21] to chemical stress is very important for the development and survival of the host. In the present study, we found that two of three tested cytochromes P450, cyp9q2, and cyp9q3 were upregulated in adult forager bees after exposed to FPF for six days, suggesting that honey bees activate the expression of specific genes rather than all genes to respond to chemical toxicity. Nevertheless, the expression of three immune genes (Abaecin, Defensin, ApidNT) and three detoxification-related genes (cyp9q1, cyp9q2, cyp9q3) had no significant difference between the exposed and control group at the larval stage.

The expression level of Defensin and ApidNT was upregulated in the new bees that emerged from the exposed larvae. The increasing expression of the genes in the immune system with exposure time would reflect the ability of bees to recover from chronic pesticide toxicity. CYP450 family members in clade CYP9 were most frequently involved in xenobiotic detoxification and evolution of the hormonal and chemosensory processes in bee species [48]. Manjon et al. unequivocally demonstrated that CPY9Q1, 2, and 3 are exclusively responsible for imidacloprid metabolism in *A. mellifera* and *Bombus terrestris* [49].

Moreover, we also investigated the effects of FPF exposure on task division in newly emerged bees and adult forager bees. In this study, there was no significant difference in Amfor and AmOctαR-1 in either exposed forager or newly emerged bees (Figure 4). Furthermore, the insulin/insulin-like signaling (IIS) pathway is an evolutionarily conserved module in the control of reproductive division of labor and foraging behavior in honey bees. An investigation into the effects of manipulated dietary regimes in early larval development on the expression of two insulin-like peptides, an insulin receptor, and two downstream components in the IIS pathway revealed caste-specific differences, particularly for the AmILP-1 and AmInR-2 encoding genes [50]. The upregulation of AmInR-2 in exposed adult bees may imply that FPF is somehow involved in the IIS pathway activity and needs further evidence for validation.

## 5. Conclusions

Altogether, our findings are in line with previous statements that FPF is more friendly to bees (no significant effect on developmental duration (larval, pupal, and total), homing, and pollen collection ability of forager bees) compared to neonicotinoid insecticides. We did find a statistically significant decrease in the survival rate of honey bees from hive colonies exposed to FPF. In addition, the qRT-PCR results suggested that the immune, detoxification, and IIS pathway related genes in newly emerged and foraging bees were affected after FPF exposure. More important synergistic and behavioral effects that can affect colony fitness should be explored in the future, considering the wide use of FPF on crops pollinated and visited by honey bees.

## Figures and Tables

**Figure 1 insects-12-00357-f001:**
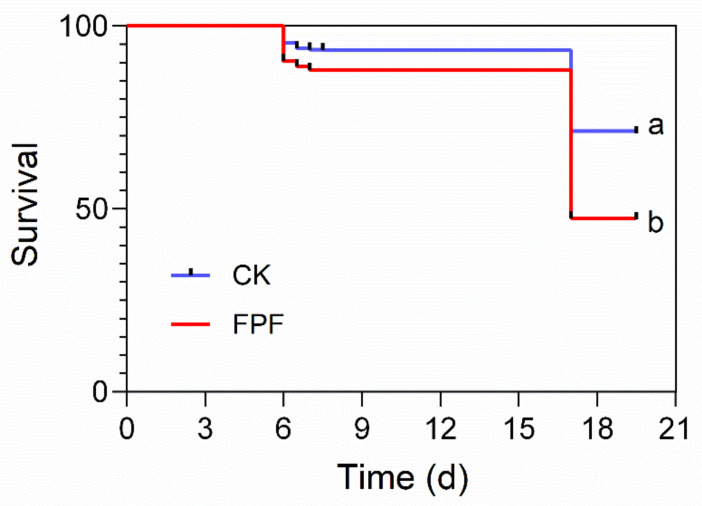
Total survival of honey bees exposed to flupyradifurone (FPF) during larval development (*n* = 4 replicates of 130 larvae/replicate). Different letters between the curve lines represent significant differences (Log Rank test, *p* < 0.001).

**Figure 2 insects-12-00357-f002:**
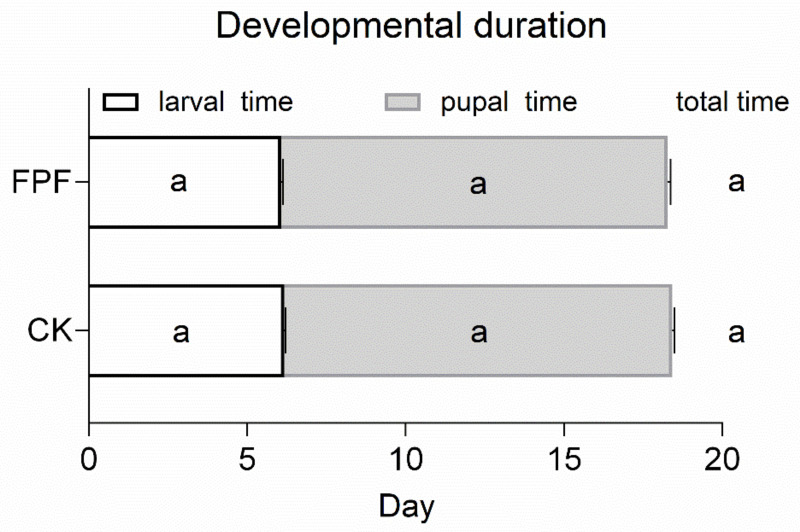
The effect of FPF on developmental duration. Values are mean ± SE (*n* = 4). Bars with the same letter present not significant differences at *p* > 0.05 by the general linear modules. Larval development time = date of pupation − hatching date. Pupal development time = emergence date − date of pupation initiation. Total development time = emergence date − hatching date.

**Figure 3 insects-12-00357-f003:**
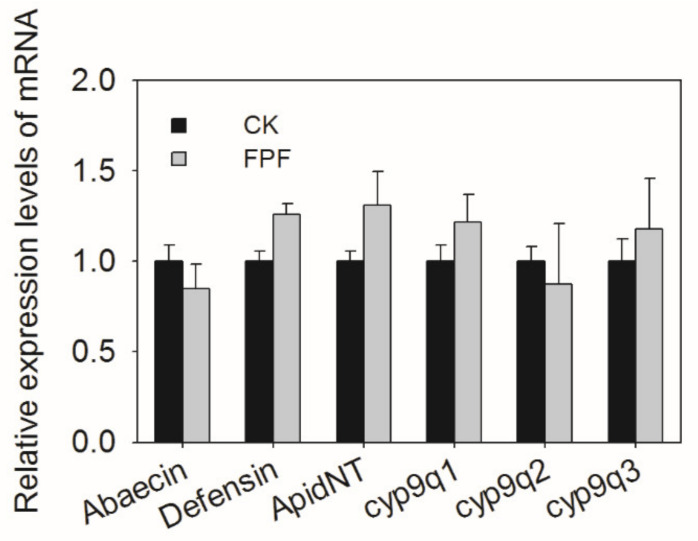
Effects of FPF on the relative expression levels of on the immune genes (Abaecin, Defensin, and ApidNT), detoxification genes (CYP9Q1, CYP9Q2, and CYP9Q3) in *A. mellifera* larvae. After exposed to 4 ppm FPF for five days, larvae were collected and total RNA was extracted. Each sample was assayed three times. Expression levels were normalized to Actin and then to the gene expression level of the CK. Values represent mean ± SEM (*n* = 4). Significant differences between FPF treatment and CK were analyzed by one-way ANOVA.

**Figure 4 insects-12-00357-f004:**
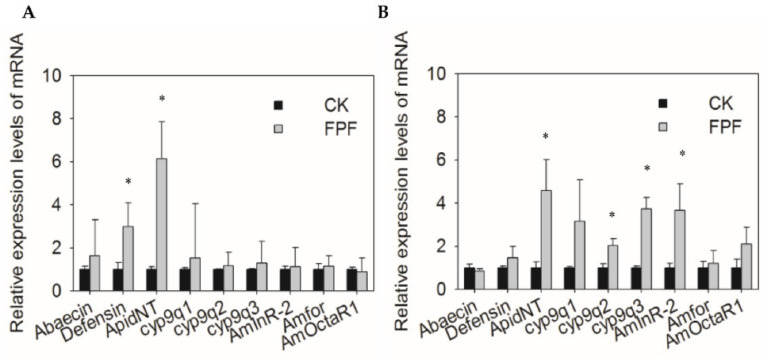
Effects of FPF on the relative expression levels of genes in newly emerged bees (**A**) and forager bees (**B**). Using four colonies run three replicates. Ten bees were randomly selected and pooled together as one biological replicate sample for gene expression analysis. Expression levels were normalized to Actin and then to the gene expression level of the CK. Values represent mean ± SEM (*n* = 4). Significant differences between FPF treatment and CK were analyzed by one-way ANOVA (* *p* < 0.05).

**Table 1 insects-12-00357-t001:** The forward and reverse primers of genes used in qRT-PCR.

Primer	Direction	Sequence 5′–3′	References
Abaecin	Forward	CAGCATTCGCATACGTACCA	[32]
	Reverse	GACCAGGAAACGTTGGAAAC	
Defensin	Forward	TGCGCTGCTAACTGTCTCAG	[32]
	Reverse	AATGGCACTTAACCGAAACG	
Apidaecin (ApidNT)	Forward	TTTTGCCTTAGCAATTCTTGTTG	[33]
	Reverse	GTAGGTCGAGTAGGCGGA TCT	
cyp9q1	Forward	TCGAGAAGTTTTTCCACCG	[34]
	Reverse	CTCTTTCCTCCTCGATTG	
cyp9q2	Forward	GATTATCGCCTATTATTACTG	[34]
	Reverse	GTTCTCCTTCCCTCTGAT	
cyp9q3	Forward	GTTCCGGGAAAATGAATC	[34]
	Reverse	GGTCAAAATGGTGGTGAC	
Amfor	Forward	CGTTTGGATCACGGAAGAAAG	[35]
	Reverse	AATACGTTGCACCGGAAGTTATATT	
AmInR-2	Forward	GGGAAGAACATCGTGAAGGA	[36]
	Reverse	CATCACGAGCAGCGTGTACT	
AmOctαR-1	Forward	GCAGGAGGAACAGCTGCGAG	[37]
	Reverse	GCCGCCTTCGTCTCCATTCG	
β-actin	Forward	TTGTATGCCAACACTGTCCTTT	[38]
	Reverse	TGGCGCGATGATCTTAATTT	

**Table 2 insects-12-00357-t002:** Mean number of foragers entering their colonies and foragers entering their colonies loaded with pollen of *A. mellifera*. Numbers are mean ± SE (standard error). Forgers entering their hives and foragers returned loaded with pollen grains at 7–9 am at days 0, 8, and 12 after FPF exposure. Same letters following the data in the same column mean no significant differences (*p* > 0.05).

Treatment	*n*	All Foragers Entering Their Colonies	Foragers Entering Loaded with Pollen
Day 0 after exposure			
Control	4	658.75 ± 206.72a	152.50 ± 34.59a
FPF	4	550.25 ± 81.87a	177.75 ± 44.57a
Day 8 after exposure			
Control	4	196.50 ± 41.12a	43.00 ± 10.51a
FPF	4	307.00 ± 45.72a	77.00 ± 22.47a
Day 12 after exposure			
Control	4	740.25 ± 298.57a	91.00 ± 45.30a
FPF	4	427.25 ± 112.73a	98.00 ± 56.08a

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
