# Peer review of "The Effects of Exposure to Flupyradifurone on Survival, Development, and Foraging Activity of Honey Bees (Apis mellifera L.) under Field Conditions"

_insects, 2021, doi:10.3390/insects12040357_

Round 1
Reviewer 1 Report
The paper by Guo et al. evaluates the effects that exposure of honeybess to FPF causes on survival, development, and foraging activity at the colony level. Given the relentless decline of pollinators populations, most of which imputable to massive use of chemicals, I think that the subject of the paper is of great interest. Moreover, the effects at the colony level, usually less evident than the direct toxicity on the individuals, are often neglected and some pesticides are considered as allegedly safe because their effects are measured only as lethality on adult bees. Thus, I think that the paper deserves to be published after several minor adjustments.
First of all, I strongly recommend a thorough English revision; I can see that the authors are not native speakers, and neither am I, but there is a number of mistakes that should be corrected. I enlist several of them below, but thare are many, therefore a language revision is seriously needed.
In the title the authors could replace “forager activity” with “foraging activity”.
Lines 28, 41: “in the future” instead of “in the further”.
Line 48: although the concern regards all species of bees and pollinators in general, I would say that more specifically the in the case of the genus Apis is especially the decline of the western honeybee, Apis mellifera, which raises the highest concern.
Line 51: rather than “debated” I would say that one of the major causes of such a decline is the massive use of pesticides in agriculture.
Lines 63-67: I would suggest to the authors to cite also other recent papers demonstrating that also allegedly safe biopesticides can cause sublethal effects at the colony levels in terms of disruption of cognitive, foraging and recognition abilities, which can undermine colony integrity and survival (see Cappa et al. 2019 Natural biocide disrupts nestmate recognition in honeybees. Scientific reports, 9, 1-10; Carlesso et al. 2020 Exposure to a biopesticide interferes with sucrose responsiveness and learning in honey bees. Scientific reports, 10, 1-12)
Lines 134-136, 184-186: the authors should invert the dates that come first (e.g. hatching date-pupation date). Furthermore, why do they use “date of pupation” (line 134) and then “date of pupation initiation”(line 135)? If it’s the same the authors should be consistent.
Line 140: use “entered” instead of “entranced”, the verb entrance (synonym to captivate) has a different meaning from the noun entrance (e.g. hive entrance).
Lines 139-141: it is not immediatly clear why the authors separated the bees that entered the hives loaded with pollen grains or all the bees the returned to the hives. I think that most of these bees should also nectar and water foragers (apart from the minority of new recruits taking the first foraging flights), so the authors could specify that the different categories of foragers collecting different resources were considered.
Line 164,167: “models” instead of “modules”.
Line 166: “that entered” or “entering” instead of simply “workers entered”.
Line 180: “pupal” instead of “papal”.
Line 189: “bees that entered hives”.
Figure 2 and Table 2. The authors should specify in the captions (and also in the statistical analysis paragraph) which statistical test they used for the comparisons and provide also the results of the test and not only the significance level.
Lines 216-218: the authors should provide the values for the test for the genes that signicantly differed in their expression.
Line 264: “chronic” instead of “chronically”.
Author Response
Dear reviewer,
Thanks for the your comments.
Please see the attachment.
Kind regards, Yi Guo
Reviewer 2 Report
Generally, I like this work very much. It brings new, interesting and important information about the influence of flupyradifurone declared by US EPA as an alternative to neonicotinoids with ’very promising safety profile when compared to other insecticides’.
Minor comments (suggestions) are inserted directly in MS (attached PDF).
However, here I give a list of some seriour shortcommings that requires explanations and/or justifications.
1) The authors wrote „colony level“ 11 times in manuscript (including title) although only four colonies per groups were used (Lines 113-114). Four colonies per group is insufficient for „colony-level“ experiment. It is very important to explain why you used only four colonies per gruop (not to me and the Editor, but also in manuscript, as it is crucial in experimental design). As I can see (looking at the nature of samples), it appears that experimental unit is not colony for any of parameters you monitored, so that is something I would recommend you to explain and specify experimental units in your experiments. Related to that, I would suggest to delete „colony level“ throughout the manuscript (including title) and, in some places, write „in field conditions“ instead. (I inserted this suggestion directly in manuscript several times with, but please make the appropriate change everywhere).
2) In connection with this info: „The experiments were conducted from August to September when the nectar flow period had already passed. Each hive consisted of six frames of adult bees...“, in that period of the year (August to September) the colonies are not strong enough, especially for monitoring developmental stages. It would be good if you may give a rationale for choosing this period of the year for the experiment. Maybe this period (August to September) is related with the treatment period of FPF in agriculture.
3) In Lines 137-141 you described how you had estimated foraging activity referring to the reference 27 (Ali, 2011). However, looking into the reference Ali (2011) I realized that your methodology for foraging activity estimation differs from that described by Ali (2011). Please, emphasize that in manuscript. For example, you may write „For estimation of foraging activity, methodology of Ali (2011) was used, but with modifications as follows: ... (describe your method)“.
4) The manuscript is written in poor English and was not checked by a holder of CPE certificate or a native speaker. Due to serious syntax and grammar defficiencies, I suggest authors to engage an experienced proofreader who is familiar with the topic of this research.
Finally, I would recommend you to include following papers in your manuscript, either in Introduction or Discussion (as they are closely related with your work, especially when it comes to the effects of agricultural pesticides on immune and detoxification genes of honey bees.
- Glavinic U, Tesovnik T, Stevanovic J, Zorc M, Cizelj I, Stanimirovic Z, Narat M (2019) Response of adult honey bees treated in larval stage with prochloraz to infection with Nosema ceranae. PeerJ 7:e6325.
- Tesovnik T, Zorc M, Ristanic M, Glavinic U, Stevanovic J, Narat M, Stanimirovic Z (2020) Exposure of honey bee larvae to thiamethoxam and its interaction with Nosema ceranae infection in adult honey bees, Environmental Pollution 256, 113443

Author Response

(The authors gave the same response as above.)

Reviewer 3 Report
The paper deals with some effects of exposure of honey bees to flupyradifurone. It provides new and relevant information and it deserves ppublication. I would be a little more cautiouswith the conclusions: surely, FPT is more friendly to bees than neonicotinoids, but the effects shown by the Authors are rather alarming.
A few minor drowbacks should be fixed:
line 3: foraging activity instead of forager activity; the whole manuscript shoud be modified accordingly
line 3. honey bees should be written as two separate words; thorughout the manuscript it is written both as two words and a single one
lines 24-25 are not clrear; lines 33-35 are much better
line 30: what is the nerve centre of sucking pests? I presume you refer to the central nervous system
line 37: you use CK group and control group aa synonyms throughout the text; I suggest you to uniform it or, at least, to state clearly that CK means control
line 41: future instead of further
line 43: I suggest to list keywords not present in the title to improve the retrieval of your paper from data banks
line 66: double space between e.g. and climbing
line 69: add space between 2018 and reference
line 74: cancel that
line 80: double space between Agency and has
line 82: what is the honey speen; may be you mean honey supers
line 87: add space between mellifera and reference
lines 90-100 deals with research carried out on Apis mellifera; the mention of Apis cerana at line 87 is misleading giving the impression that the cited research deals with it
lines 112-113: I presume that the experimental colonies are queenless; if so you shoud state it
line 119: following instead of followed
line 140: entering instead of entranced or entered or introduced throghot the manuscript
line 143: I do not understan what respectively refers to
lines 164-167: in my espenrience, log-transpormed data tend to hide differences; I would ha carried out a chi-square test on the raw data and suggest you to try it
line 180: pupal instead of papal
line 184: add differences afte significant
line 203: looking at figure 3, I would say that defensin show a significant difference; I have no reason to suspect that your calculations are wrong, but I suggest yo to report the exact P values and not simply note P > 0.05
line 241: exposure instead of exposed
line 265: reduced instead of reducing
Author Response

(The authors gave the same response as above.)
